# More than Just a Shell: Indehiscent Fruits Drive Drought-Tolerant Germination in Invasive *Lepidium* Species

**DOI:** 10.3390/plants14101517

**Published:** 2025-05-19

**Authors:** Said Mohammed, Klaus Mummenhoff

**Affiliations:** 1Department of Biology, College of Natural and Computational Sciences, Debre Berhan University, Debre Berhan 445, Ethiopia; 2Department of Biology, University of Osnabrück, Barbarastraße 11, D-49076 Osnabrück, Germany; kmummenh@uni-osnabrueck.de

**Keywords:** distribution, drought stress, invasive species, *Lepidium*, seed ecology

## Abstract

This study aims to assess the drought stress tolerance of germinating seeds of the invasive indehiscent fruit-producing *Lepidium* species, specifically *Lepidium appelianum*, *Lepidium draba*, and the invasive dehiscent fruit-producing *L. campestre*. Drought stress tolerance experiments were conducted using various concentrations of polyethylene glycol (PEG) following standard protocols. The results revealed that seeds/fruits of *L. appelianum* and *L. draba* exhibited significantly higher drought stress tolerance compared to seeds of *L. campestre*. Fresh seeds and fruits of *L. appelianum* were capable of germinating under various drought stress treatments, while fresh and after-ripened seeds and fruits of *L. draba* germinated in every condition except for −0.8 MPa. Conversely, *L. campestre* fresh seeds did not germinate under the most severe drought stress conditions (−0.6 and −0.8 MPa). It is crucial to note that fresh fruits of *L. draba* displayed pericarp-mediated chemical dormancy, while fresh seeds of *L. campestre* demonstrated physiological dormancy. However, fresh seeds and fruits of *L. appelianum* did not exhibit any dormancy. This study suggests that germinating seeds and fruits of *L. appelianum* demonstrate the strongest tolerance to drought stress, while *L. draba* exhibits moderate tolerance. On the contrary, *L. campestre* seeds display the least tolerance to drought stress. The differences in drought stress tolerance among the studied *Lepidium* species reflect the climatic facets in their native distribution areas. Given the potential high invasiveness associated with the drought stress tolerance of *L. appelianum* and *L. draba*, it is imperative to develop special control strategies to manage these invasive species in the face of future climate change.

## 1. Introduction

Biological invasions pose a significant threat to global biodiversity, especially as a result of the influences of globalization and climate change [1]. Invasive plant species are recognized as a major driver of environmental change, leading to detrimental effects on biodiversity, ecosystem services, environmental quality, and human health [2,3]. The economic losses resulting from plant invasions are substantial and have far-reaching effects on agriculture, horticulture, and natural ecosystems [4,5]. Successful invasive species often exhibit traits that contribute to their ability to outcompete native species, notably adaptability to various environmental conditions, including drought stress [2]. Specifically, invasive weeds tend to demonstrate higher tolerance to drought stress compared to crops [6,7], leading to reduced crop productivity and increased weed infestation during drought seasons [8,9]. Moreover, seed traits play a crucial role in a species’ invasiveness, influencing its ability to cope with drought stress conditions [10,11]. Species that demonstrate adaptability to a wide range of environmental conditions are more likely to be successful invaders and exhibit higher invasiveness [12,13]. Our study focuses on closely related invasive indehiscent fruit-producing *L. appelianum* and *L. draba* and dehiscent fruit-producing *L. campestre*. *L. appelianum* Al-Shehbaz (syn. *Cardaria pubescens* (C.A. Meyer) Jarmolenko, *Hymenophysa pubescens* C.A. Meyer) is also known as globe-podded hoary cress and *L. draba* L. (syn. *Cardaria draba* (L.) Desvaux) is known as heart-podded hoary cress [14,15,16]. *Lepidium campestre* (L.) W.T. Aiton is commonly known as field pepperwort [17].

*L. appelianum* is native to Central Asia, including Kazakhstan and Uzbekistan, where it experiences a harsh continental climate with hot, dry summers and cold winters [18]. This invasive weed has also distributed to Argentina, Canada, and the USA [18,19,20]. Similarly, *L. draba*, is native to the Middle East and Mediterranean regions, particularly the Balkan Peninsula, Turkey, and Iran [21]. This species has also become invasive in Europe [22], Australia, the USA, and Canada [23,24,25]. In contrast, the invasive *L. campestre* is native to Europe, where it experiences a more humid climate and sufficient rain supply [26,27].

According to Mohammed et al. [28], freshly harvested seeds and fruits of *L. appelianum* are non-dormant, while *L. draba* is characterized by pericarp-mediated chemical dormancy, attributed to the inhibitory effect of the abscisic acid (ABA) residing in the fresh pericarp tissues. Similarly, Mohammed et al. [29] reported that *L. campestre* demonstrates physiological seed dormancy, and this is supported by Partzsch [27]. It is important to mention that after-ripening releases dormancy in both *L. draba* and *L. campestre* [28,29,30]. These studies were conducted under optimal germination conditions for the respective species [28,29,30]. However, the seed and fruit germination ecology under drought stress conditions remain largely unknown for these species. The dispersal units of *L. appelianum* and *L. draba* consist of the entire indehiscent fruit, which allows for germination directly from the fruit itself [26,28,29,30,31,32]. In contrast, *L. campestre* relies on seeds released from dehiscent fruits as its dispersal units [26,27,29,32], as shown in Figure 1. A noteworthy common characteristic between these species is their seeds’ ability to become mucilaginous upon wetting [31,32]. Various methods have been used to assess species tolerance to drought stress during seed germination [7,32,33]. One such approach is the application of polyethylene glycol (PEG) to seeds at different concentrations [34,35]. PEG applications have been shown to be effective in evaluating drought stress tolerance in many plant species [7,34]. PEG compounds have been used to simulate osmotic stress effects in petri dishes (in vitro) and ensure uniform water potential for plants throughout the experimental period [36,37]. The water stress induced by polyethylene glycol was similar to that in the field, allowing an assessment of seed germination under drought stress [13,38]. Therefore, this study is focused on examining the effects of polyethylene glycol (PEG)-induced drought stress tolerance on seed and fruit germination of three invasive weed species, namely *L. appelianum*, *L. draba*, and *L. campestre*. Understanding the drought stress tolerance of these species could provide valuable insights for developing strategies to control invasive species, thereby enhancing agricultural productivity and biodiversity. Based on the native distribution range of the studied species, it can be hypothesized that *L. appelianum* and *L. draba* demonstrate greater drought stress tolerance compared to *L. campestre*. Given the significant impact of invasive weeds on agriculture and wildlife diversity, understanding the ecological significance of dehiscent and indehiscent fruits is crucial for enhancing agricultural productivity and conserving biodiversity. Insights into the germination behaviors and adaptive strategies of these diaspores can inform effective management plans aimed at mitigating their harmful effects on ecosystems and farming practices.

## 2. Results

### 2.1. Indehiscent-Fruited L. appelianum and L. draba Exhibit High Drought Tolerance

Our study found that fresh fruits of *L. appelianum* achieved a high germination rate (88.3 ± 1.6%) without drought stress and showed no dormancy related to its seeds or fruits. Germination rates under varying drought conditions were 83.3 ± 0.8%, 80 ± 1.15%, 52 ± 1.6%, and 38 ± 1.7% at −0.2, −0.4, −0.6, and −0.8 MPa, respectively, indicating a decline in germination with increased drought stress. Significant differences were found in germination rates between 0 MPa and −0.4 MPa (*p* = 0.005), −0.6 MPa (*p* < 0.001), and −0.8 MPa (*p* < 0.001), while differences between 0 MPa and −0.2 MPa (*p* = 0.095) and −0.2 MPa and −0.4 MPa (*p* = 0.37) were not significant.

Furthermore, fresh seeds of *L. appelianum* had a high germination rate of 86.67 ± 1.6%, confirming the absence of seed dormancy. Under non-drought stress conditions (0 MPa), germination exceeded 85%, but decreased to 82.3 ± 1.4%, 78.3 ± 1.6%, 52.7 ± 1.8%, and 36 ± 1.2% at −0.2, −0.4, −0.6, and −0.8 MPa, respectively (Figure 2 and Appendix A).

When subjected to non-drought stress conditions, fresh fruits of *L. draba*, which contain seeds enclosed within the freshly matured pericarp, displayed 56 ± 2.3% germination (Figure 3). This indicates that >40% of the fresh fruits of *L. draba* did not germinate, suggesting that the species exhibits dormancy. Under drought stress conditions of −0.2, −0.4, −0.6, and −0.8 MPa, the germination percentages for fresh fruits of *L. draba* were 42.3 ± 1.4%, 17.3 ± 1.3%, 14 ± 1.1%, and 0 ± 0.0%, respectively. None of the seeds germinated under −0.8 MPa.

Moreover, in this study, we found that after-ripened fruits of *L. draba* exhibited a significantly (*p* < 0.001) higher germination percentage (86.6 ± 1.6%) compared to freshly matured *L. draba* fruits (56 ± 2.3%). Our experiments resulted in germination percentages of 53.3 ± 3.7%, 47.3 ± 4.3%, 20 ± 1.5%, and 0 ± 0.0% under −0.2, −0.4, −0.6, and −0.8 MPa, respectively (Figure 3) in *L. draba’s* after-ripened fruit germination. The difference in germination percentage between 0 MPa with −0.2, −0.4, −0.6, and −0.8 MPa was found to be significant (*p* < 0.001, Appendix A) in *L. draba’s* after-ripened fruit germination. After-ripened fruits of *L. draba* exhibited significantly higher germination percentages (*p* < 0.001; Appendix A) at 0, −0.2, −0.4, −0.6, and −0.8 MPa compared to freshly harvested *L. draba* fruits.

Furthermore, the germination percentage of *L. draba* fresh seeds (seeds manually released from the pericarp) indicates that the seeds of *L. draba* are non-dormant. Under non-drought stress (0 MPa) conditions, *L. draba* fresh seeds germinated to 86.6 ± 1.6% (Figure 3). However, under −0.2, −0.4, −0.6, and −0.8 MPa, the germination percentages were 53.6 ± 1.2%, 45.3 ± 0.6%, 8.6 ± 1.7%, and 0 ± 0.0%, respectively (Figure 3). The difference in germination percentages between 0 MPa with −0.2, −0.4, −0.6, and −0.8 MPa was found to be statistically significant (*p* < 0.001, Appendix A) in *L. draba’s* fresh seed germination. *L. draba* fresh seeds exhibited significantly high germination percentages at 0, −0.2, −0.4, −0.6, and −0.8 MPa compared to *L. draba* fresh fruits.

### 2.2. Dehiscent Fruit-Producing L. campestre Exhibit Reduced Drought Stress Tolerance Compared to the Indehiscent Fruit-Producing L. appelianum and L. draba

Freshly matured seeds of *L. campestre* showed a lower germination percentage (34.3 ± 1.4%) and the species could demonstrate dormancy. Furthermore, the germination percentage decreased to 18 ± 1.1%, 6 ± 1.2%, 0 ± 0.0%, and 0 ± 0.0% under −0.2, −0.4, −0.6, and −0.8 MPa, respectively (Figure 4). Notably, no germination was observed at −0.6 and −0.8 MPa. The difference in germination percentage between 0 MPa with −0.2, −0.4, −0.6, and −0.8 MPa was found to be statistically significant (*p* < 0.001, Appendix A). Fresh seeds of *L. campestre* exhibited significantly lower germination percentages in all measured drought stress treatments compared to *L. appelianum* and *L. draba* (*p* < 0.001, Appendix A).

Moreover, we observed that after-ripened *L. campestre* seeds displayed a higher germination percentage (86.6 ± 1.6%) than fresh seeds of *L. campestre* (34.3 ± 1.4%) under non-drought stress conditions (Figure 4). Additionally, we noted germination percentages of 46 ± 1.4, 34.6 ± 1.7, 9.3 ± 3.5, and 0 ± 0.0 under −0.2, −0.4, −0.6, and −0.8 MPa, respectively (Figure 4). The difference in germination percentage between 0 MPa with −0.2, −0.4, −0.6, and −0.8 MPa was found to be statistically significant (*p* < 0.001, Appendix A). After-ripened seeds of *L. campestre* exhibited significantly higher germination percentages (*p* < 0.001, Appendix A) at 0, −0.2, −0.4, −0.6, and −0.8 MPa compared to fresh seeds of *L. campestre*.

### 2.3. Dehiscent vs. Indehiscent: L. appelianum and L. draba’s Indehiscent Fruits Could Benefit from Future Climate Change Relative to Dehiscent Fruit-Producing L. campestre

The results indicated that both *L. appelianum* and *L. draba*, which produce indehiscent fruits, exhibited a significantly higher germination percentage (*p* < 0.001, Appendix A) compared to *L. campestre*, which produces dehiscent fruits. Across the drought stress treatments, *L. appelianum* and *L. draba* (not at −0.8 MPa) consistently showed a significantly higher germination percentage (*p* < 0.001, Appendix A) than *L. campestre* (Figure 5). Specifically, seeds and fruits of *L. appelianum* germinated in various drought stress treatments, while *L. draba* germinated in every treatment except for −0.8 MPa. In contrast, *L. campestre* seeds did not germinate at −0.6 and −0.8 MPa (Figure 5).

## 3. Discussion

### 3.1. Indehiscent Fruits Drive Enhanced Drought Tolerance and Invasive Potential in *Lepidium*

The germination responses of the three different species, namely *L. appelianum*, *L. draba*, and *L. campestre*, to drought stress highlight significant variances in drought tolerance related to their fruit and dispersal types. *L. appelianum* and *L. draba*, both of which produce indehiscent fruits, demonstrated robust germination rates even under varying drought conditions, with *L. appelianum* achieving an impressive 88.3% germination without drought stress and still maintaining 38% viability at considerable drought stress (−0.8 MPa). This adaptability in survival and reproduction, particularly in environments with limited water availability, provides *L. appelianum* and *L. draba* with a competitive edge over *L. campestre*, which exhibited severe reductions in germination success under similar drought stress. The capacity of indehiscent fruits to retain moisture and possibly enhance seedling establishment allows these species to spread effectively in arid regions, raising concerns about their invasiveness and the ecological implications for the native flora. Moreover, the findings align with the existing literature highlighting the crucial role of seed and fruit morphology in shaping drought resilience. For instance, Kintl et al. [41] found that morphological traits significantly influence germination rates under drought stress across different clover species. This reinforces the idea that the indehiscent fruits of *L. appelianum* and *L. draba* provide not only a survival advantage but also enhance their capacity to outcompete native species, facilitating their establishment in disturbed environments. Such behaviour, driven by the evolutionary advantages conferred by fruit type, highlights the importance of understanding interspecies dynamics in the context of climate change and resource scarcity.

Other studies emphasize the role of genetic traits and environmental factors in drought tolerance, highlighting the invasive potential of indehiscent fruit-bearing species, like *L. appelianum* and *L. draba*. Mehmandar et al. [42] found that drought stress negatively impacted the physiological traits of Iranian melon genotypes, mirroring our findings of reduced germination rates for *L. campestre*. In contrast, *L. appelianum* and *L. draba* exhibited higher germination rates under drought conditions, suggesting that their indehiscent fruit morphology aids in seed retention and moisture conservation, enabling these species to thrive and compete effectively against *L. campestre*. Additionally, Li et al. [43] and Licaj et al. [44] highlighted that environmental factors, like temperature and photoperiod, are crucial for drought resistance, allowing *L. appelianum* and *L. draba* to adapt to changing climates. Furthermore, key genes linked to drought tolerance, as noted by Lu et al. [45], may facilitate breeding efforts for these invasive species. Furthermore, the dormancy patterns exhibited by *L. draba* further complicate the implications of these findings, as the species displayed significant dormancy in its fresh fruits, influencing initial germination rates. However, after-ripening improved germination significantly, indicating that environmental stress cues can enhance the viability of seeds over time. Such mechanisms suggest that *L. draba* and *L. appelianum* are well-equipped to thrive under fluctuating climate conditions, effectively positioning them as models for studying invasive traits and resilience strategies. Understanding the distinctions of fruit type in relation to drought resilience not only contributes to our knowledge of plant ecology but also raises critical questions about the management of invasive species in increasingly arid landscapes.

### 3.2. Implications of Indehiscent Fruit Evolution for Drought Stress Adaptation in *Lepidium*

Throughout the evolutionary history of flowering plants, a range of fruit shapes has emerged, many of which play a crucial role in seed dispersal and germination [46,47]. Fruits can be classified as dehiscent fruits, which open soon after maturation to release seeds, or indehiscent fruits, which remain closed and serve as the means of dispersal [48]. Dehiscent fruits are the predominant type and are believed to represent the ancestral form within the Brassicaceae family [49,50]. Molecular studies indicate that indehiscent fruits have evolved independently multiple times from their dehiscent counterparts [49,50]. The closely related species, such as *L. appelianum* and *L. draba*, produce non-fleshy, indehiscent fruits (Al-Shehbaz, [31]; Figure 1), where the fruit valves enclose the two seeds during dispersal (Mohammed et al. [28]; Figure 1). In these cases, the entire fruit functions as the unit of dispersal [32,50]; Figure 1. The shift from dehiscent to indehiscent in fruit could potentially be understood within the context of evolving species demonstrating increased drought stress tolerance. Our findings strongly support this hypothesis, as we found that *L. appelianum* and *L. draba*, which produce indehiscent fruit, exhibit greater drought stress tolerance in comparison to *L. campestre*, which produces dehiscent fruit.

### 3.3. Native Distribution Areas of Indehiscent-Fruited Lepidium Species Could Suggest Drought Stress Tolerance

*L. appelianum* and *L. draba*, both native to regions with extreme continental, arid, and semiarid climates, demonstrate an inherent adaptation to high drought stress during germination [18,21]. *L. appelianum* is native to central Asia, encompassing arid regions in Kazakhstan, Uzbekistan, Turkmenistan, northern Iran, and Afghanistan, characterized by long, hot, dry summers and cold, dry winters according to Francis and Warwick [18]. Similarly, *L. draba* is believed to be native to central Asia and Siberia, including the Balkan Peninsula, Georgia, Armenia, Azerbaijan, Turkmenistan, Kazakhstan, southern Russia, Turkey, Israel, Syria, Iraq, and Iran [18,21], i.e., regions with a Mediterranean and continentally influenced climate, featuring warm summers but very cold and dry winters [18,21]. In contrast, *L. campestre*, also known as field cress (field pepperweed), is native to Europe [27,51], including countries in the Nordic region, with a broad distribution across Europe, and is found in the humid/humid nemoral zone with reliable and sufficient rain supply [17,27]. These distinctive native habitats and distributions highlight the inherent adaptability of *L. appelianum* and *L. draba* to regions with high drought stress, further indicating their potential for high drought stress tolerance compared to *L. campestre*, which is more adapted to humid environments.

The widespread distribution of *L. appelianum* and *L. draba* across continents, except Antarctica, presents a significant ecological and agricultural concern [19,51]. *L. appelianum* has been reported in various countries, including Argentina, Canada, and the USA [14,18], but is not listed among established species in Europe [18]. Conversely, *L. draba* is most prevalent in arable land in Europe [18] and other parts of the world [18] and has become established as an agricultural weed in Canada, Australia, and the United States [18,52]. In the USA, Canada, and Australia, *L. appelianum* and *L. draba* are classified as a noxious and invasive weed, posing challenges in terms of control and eradication [18]. On the other hand, the distribution patterns of *L. campestre* suggest its invasive nature in regions with sufficient rainfall [26]. These findings suggest the importance of continued monitoring and management strategies to mitigate the impact of the indehiscent fruit-producing invasive *Lepidium* species on ecosystems and agricultural practices, further highlighting potential for their high drought stress tolerance and adaptability across diverse geographical regions.

### 3.4. Global Ranking of the Invasiveness of Indehiscent Fruit-Producing Lepidium Species Indicates Potential Drought Stress Tolerance

It is evident that the invasiveness of *L. appelianum* and *L. draba* could be related to their drought stress tolerance during seed germination. These species have been identified as significant noxious weeds, ranking 8th out of 45 in multiple noxious weed lists across the United States, Australia, and Canada, with appearances on 17 of 48 lists, as documented by Skinner et al. [53], and the USDA [24]. They have been observed to infest agricultural land, pastures, riparian areas, and waste areas in these regions, as supported by research from Mulligan [54]. On the other hand, *L. campestre* has been noted to be invasive in areas within its distribution range that receive sufficient rainfall, as highlighted by Geleta et al. [17] and the Montana Natural Heritage Program [51]. This suggests that the invasiveness of *L. campestre* may be more restricted by specific environmental conditions, indicating a potential difference in the ecological impact and management strategies for this species compared to *L. appelianum* and *L. draba*.

### 3.5. Understanding the Link: How Indehiscent Fruits Facilitate Drought Stress Tolerance and Contribute to Invasiveness

The heart-podded hoary cress (*L. draba*) demonstrates a notable capacity for germination and root growth under challenging osmotic and saline conditions, as highlighted by research from Francis and Warwick [18]. Their study revealed high germination rates even at higher negative low osmotic potentials and showed that salinity levels up to an electrical conductivity of 12 dS/m had no adverse effects on germination or root growth [55]. This resilience to osmotic and saline stress indicates the adaptability of *L. draba* to diverse environmental conditions, including dry and saline habitats. Additionally, the mucilaginous nature of *L. draba* and *L. appelianum* seeds, as reported by Mummenhoff et al. [49] and Mohammed and Mummenhoff [34], facilitates adhesion to the soil surface and supports germination under low moisture conditions, presenting a potential adaptation to arid environments. This contrasts with the germination strategy of *L. campestre*, whose seeds absorb and retain much water for seed germination [34,56]. These findings suggest that *L. draba* and *L. appelianum* may possess a competitive advantage in establishing themselves under varying environmental conditions, including arid regions. The restricted invasiveness of *L. campestre* to wet and moist habitats [26] further supports the notion that different *Lepidium* species exhibit distinct ecological preferences. Understanding these species-specific responses to environmental stressors provides valuable insights into their ability to thrive and potentially dominate in particular habitats, further indicating their potential for high drought stress tolerance and adaptability.

### 3.6. The Broad Ecological Relevance of PEG in Drought Stress Simulation

Polyethylene glycol (*PEG*) has emerged as an essential compound in ecological research for simulating drought stress conditions during the critical stages of seed germination and seedling growth [7]. By inducing controlled osmotic stress in laboratory settings, PEG allows researchers to maintain consistent water potential while observing how various plant species respond to water scarcity [13]. This capability is vital for understanding the impacts of drought on seed vigor, germination rates, and overall seedling health. Additionally, *PEG* facilitates the exploration of plant physiological and biochemical responses, which can be crucial for developing resilient crop varieties and enhancing agricultural productivity in an era marked by climate variability [7]. Beyond its agricultural applications, *PEG* in the context of inducing controlled drought stress plays a significant role in studying the ecological dynamics of invasive plant species [34]. By examining how controlled drought stress via *PEG* influences seed traits and germination strategies, researchers can uncover the mechanisms that enable certain species to thrive under water-limiting conditions [36]. Such insights are critical for devising effective management strategies aimed at controlling the spread of invasive species and protecting native ecosystems [37]. Furthermore, *PEG*’s simulation of drought conditions has proven beneficial in elucidating the interactions between seed mucilage, water absorption, and seed dispersal, contributing to a deeper understanding of plant survival strategies in increasingly unpredictable environments [57]. Through these multifaceted studies, PEG serves as both a vital research tool and a key component in addressing the ecological challenges posed by drought stress.

## 4. Materials and Methods

### 4.1. Seed Sources

Seed sources for this study included mature fruits of *L. appelianum* Al-Shehbaz from 42.8 N, 69.9 E (Baydibek District, Kazakhstan), mature fruits of *L. draba* (L) from 39.5 N, 26.9 E (Burhaniye, Turkey), and mature seeds of *L. campestre* (L) W.T. Aiton from 56.8 N, 12.9 E (SJömossevägen, Halmstads kommun, Sweden). These seeds and fruits were mass propagated and collected from plants cultivated in the Botanical Garden of the University of Osnabrück, Germany. The fresh status of the seeds and fruits was maintained by storing them at −20 °C until the experiments were initiated, following a protocol described by Baskin and Baskin [58] and Mohammed et al. [28].

### 4.2. Microscopy of Lepidium Seed and Fruit Morphology and Germinating Units

The fruit morphology and germinating units of three *Lepidium* species were examined, including indehiscent fruit-producing species, namely *L. appelianum* and *L. draba*, as well as a dehiscent fruit-producing species, namely *L. campestre*. This analysis was quantified using a Leica M165 FC Fluorescence Classic stereomicroscope (Leica Microsystems, Heerbrugg, Switzerland) following the methodologies outlined by Mohammed et al. [28].

### 4.3. Seed and Fruit Material: Collection and Processing

Fresh mature seeds and fruits were collected directly from the mother plants immediately upon maturation. These freshly collected samples were air-dried at room temperature for seven days. The germination status of the seeds and fruits was assessed following the methods outlined by Baskin and Baskin [58] and Mohammed et al. [28] and stored in paper bags sealed within aluminum bags at −20 °C until the experiments were initiated. In contrast, after-ripened seeds and fruits were obtained by storing the fresh mature samples under laboratory conditions (25 ± 2 °C, 51% relative humidity) for four months, as per the protocol established by Mohammed et al. [28].

### 4.4. Dormancy Characterization and Diaspore Types of the Three Lepidium Species

Previous studies have demonstrated that fresh *L. draba* fruits exhibit pericarp-mediated chemical dormancy [28], while fresh seeds of *L. campestre* exhibit non-deep physiological dormancy [29]. In contrast, fresh fruits and seeds of *L. appelianum* are classified as non-dormant [28]. The dormancy of *L. draba* seeds enclosed within indehiscent fruits and *L. campestre* seeds were effectively released after 16 weeks of after-ripening [28,29]. In this study, we utilized the following plant materials: (A) fresh seeds and fruits of *L. appelianum*, (B) fresh seeds, fresh fruits, and after-ripened fruits of *L. draba*, and (C) both fresh and after-ripened seeds of *L. campestre*.

### 4.5. Dormancy Releasing Treatments

To release the dormancy of *L. draba* fresh fruits and *L. campestre* fresh seeds, the diaspores were stored under controlled laboratory conditions (25 ± 2 °C, 51% relative humidity) for a duration of 16 weeks, following the protocols established by Mohammed et al. [28] and Mohammed et al. [29].

### 4.6. Drought Stress Experiments

The germination responses to water potential of the isolated seeds and seeds enclosed within the pericarp were determined by incubating the diaspores for a 12/12 h photoperiod at the optimum temperature for the species (23–25 °C for *L. appelianum* and *L. draba*; 16–20 °C for *L. campestre*) as described by Mohammed et al. [28] and Mohammed et al. [29]. Solutions of polyethylene glycol (PEG) 8000 (Sigma-Aldrich, Germany) with water potentials of 0, −0.2, −0.4, −0.6, and −0.8 MPa were prepared using the method developed by Michel [40]. For each treatment, three replicates of 25 seeds/fruits were placed in 9 cm-diameter Petri dishes on filter paper and moistened with 4 mL of distilled water (control) or different concentrations of PEG solutions. The Petri dishes were sealed with Parafilm to reduce evaporation. To ensure relatively constant water potential during the germination period, the seeds were transferred to a new filter paper with fresh PEG solution every two days. The number of germinated seeds was recorded daily, and the germination tests ran for 4 weeks under a 12/12 h light regime, with white light of approximately 100 µmol m^−2^ s^−1^. Visible protrusion of the radicle was recorded as the completion of germination [59,60].

### 4.7. Data Analysis

One-way ANOVA was conducted to analyze the association between the drought stress treatments within and between the species. Data were analyzed using a one-way ANOVA to assess differences among groups, followed by post hoc comparisons using Tukey’s honest significant difference test. The assumption of homogeneity of variances was evaluated to ensure the validity of the results. The rejection threshold for all analyses was *p* < 0.05. Probit regression analysis was performed to determine the best fit between the actual and predicted germination values. The analysis processed the data to identify the maximum germination value at the earliest occurrence based on the observed germination values. The results were then visually represented using R version 4.3.2 (The R Foundation for Statistical Computing, Vienna, Austria).

## 5. Conclusions

This study has successfully assessed the drought stress tolerance of germinating seeds and fruits from three *Lepidium* species, namely *L. appelianum*, *L. draba*, and *L. campestre*. The results demonstrate that *L. appelianum* exhibits the highest tolerance to drought, germinating effectively across all drought stress treatments. *Lepidium draba* shows moderate tolerance, with limited germination under the most severe conditions, while *L. campestre* failed to germinate in the harsher drought environments. These findings highlight significant differences in drought tolerance, which may be linked to the climatic conditions of each species’ native distribution. The pronounced drought tolerance of *L. appelianum* and *L. draba* raises important ecological concerns regarding their potential invasiveness, particularly in the context of climate change. Given their ability to thrive under drought conditions, it is critical to formulate targeted management strategies to control these invasive species effectively. For future research, it is recommended to investigate the underlying physiological mechanisms that contribute to their drought tolerance and to assess their interactions with native plant species. Such studies will provide deeper insights into the ecological impacts of these invasive *Lepidium* species and aid in developing effective conservation and management practices.

## Figures and Tables

**Figure 1 plants-14-01517-f001:**
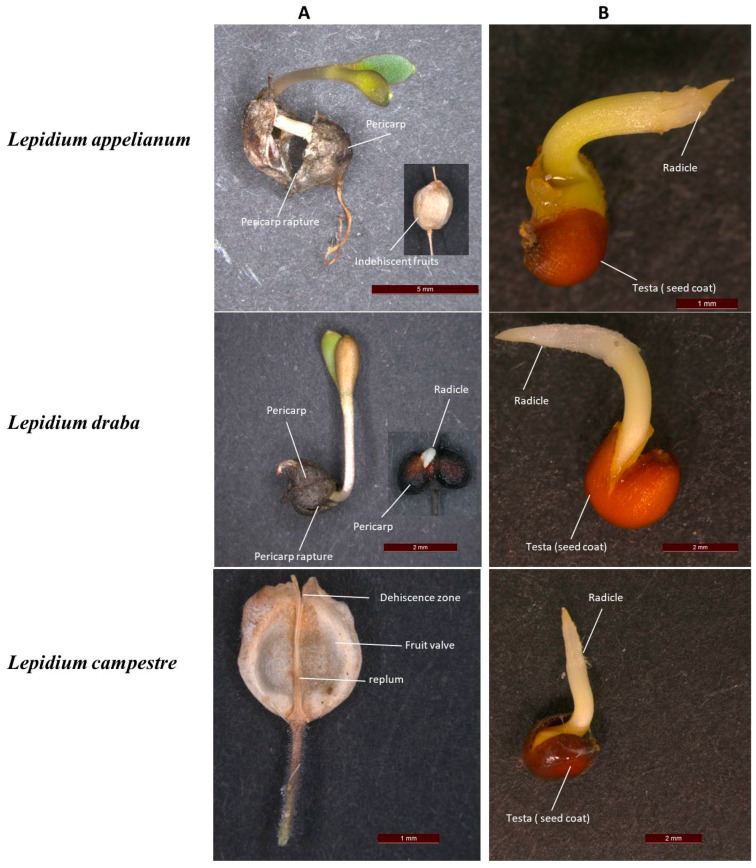
Images showing the germination of (**A**) *L. appelianum* and *L. draba* seeds within the fruit, which is the true dispersal unit of the species, and the dehiscent fruit of *L. campestre*. (**B**) Manually isolated seeds of *L. appelianum* and *L. draba* and the germinating seeds of *L. campestre*, which is a true dispersal unit of the species.

**Figure 2 plants-14-01517-f002:**
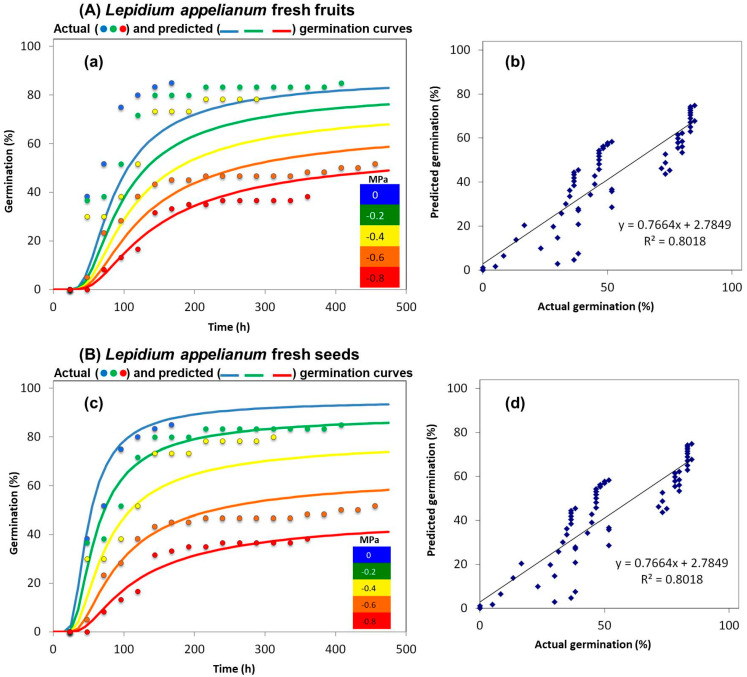
The germination percentages of *L. appelianum* fresh fruits and fresh manually isolated seeds under various drought stress conditions. (**A**) Fresh fruits: (**a**) illustrates the germination percentage at 0 (control), −0.2, −0.4, −0.6, and −0.8 MPa; (**b**) compares the correlation between actual and predicted germination percentage, with the linear relationship indicated by the R^2^ values. (**B**) Fresh fruits: (**c**) shows the germination percentage at 0 (control), −0.2, −0.4, −0.6, and −0.8 MPa, while (**d**) compares the correlation between actual and predicted germination percentage, indicating the linear relationship with the R^2^ values. Polyethylene glycol 8000 (Sigma-Aldrich, Darmstadt, Germany) solutions were prepared using a method described by Michel [39], and the germination percentages were determined following the technique outlined by Alvarado and Bradford [40]. Probit regression analysis was performed to determine the best fit between the actual and predicted germination values. The analysis processes the data to identify the maximum germination value at the earliest instance (based on the actual germination values). Each drought treatment had *n* = 3 × 25, and the germination test was conducted over a 4-week period. The actual and predicted germination percentages show a strong resemblance when the R^2^ value is closer to 1. Germination was recorded when the radicle protrusion (2 mm) became apparent.

**Figure 3 plants-14-01517-f003:**
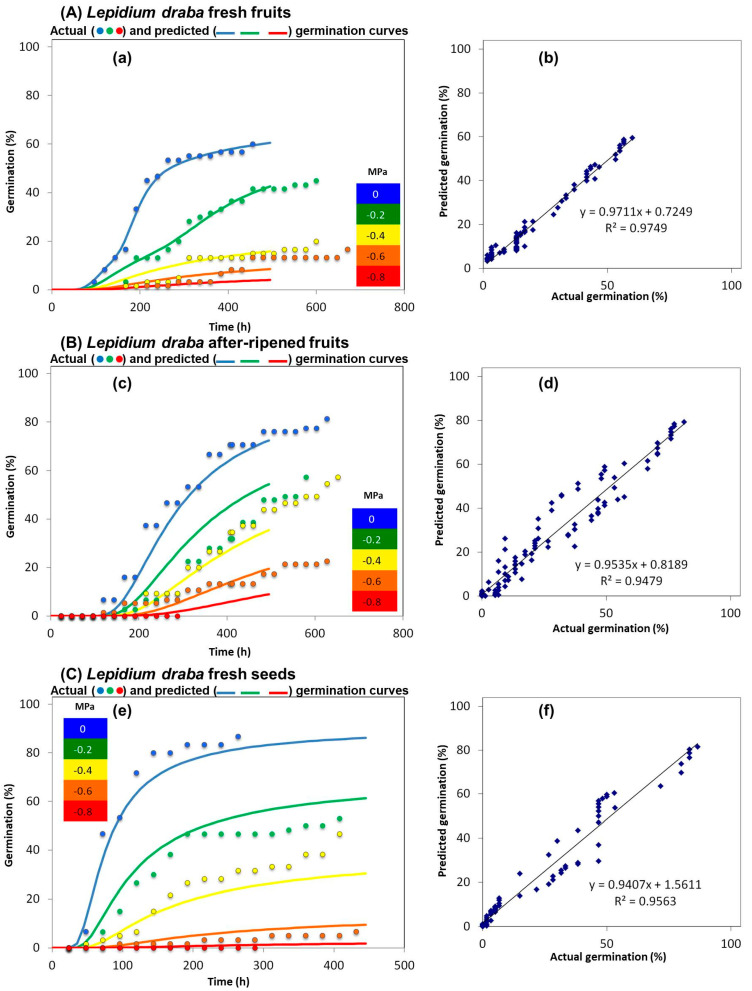
Germination percentage of *L. draba* fresh fruits, after-ripened fruits, and fresh seeds under various levels of drought stress. (**A**) Fresh fruits: (**a**) illustrates the germination percentage at 0 (control), −0.2, −0.4, −0.6, and −0.8 MPa, while (**b**) compares the correlation between actual and predicted germination percentages. (**B**) After-ripened fruits: (**c**) displays the germination percentage at different levels of drought stress (−0.2, −0.4, −0.6, and −0.8 MPa) compared to the control (0 MPa), while (**d**) compares the correlation between actual and predicted germination percentages. (**C**) Fresh seeds: (**e**) illustrates the germination percentage of *L. draba* fresh seeds at 0 (control), −0.2, −0.4, −0.6, and −0.8 MPa, and (**f**) presents the correlation between actual and predicted germination percentages. The linear relationship between the actual and predicted germination values is depicted using R^2^ values. The preparation of polyethylene glycol 8000 (Sigma-Aldrich, Germany) solutions followed a method described by Michel [39], and the germination percentages were determined a method outlined by Alvarado and Bradford [40]. Probit regression analysis was performed to determine the best fit between the actual and predicted germination values. The analysis processes the data to identify the maximum germination value at the earliest instance (based on the actual germination values). Each drought treatment consisted of *n* = 3 × 25, and the germination test was conducted over a 4-week period. The resemblance between the actual and predicted germination percentages is notable when the R^2^ value approaches 1. Germination was recorded when the radicle protrusion (2 mm) became apparent.

**Figure 4 plants-14-01517-f004:**
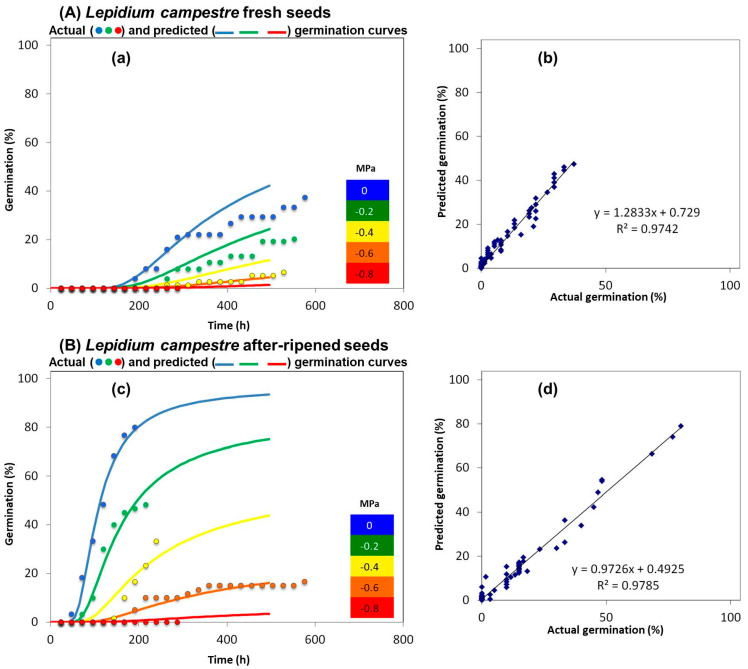
Germination percentage of *L. campestre* fresh seeds and after-ripened seeds under varying levels of drought stress conditions. (**A**) Fresh seeds: (**a**) the germination percentage of *L. campestre* fresh seeds is shown at 0 (control), −0.2, −0.4, −0.6, and −0.8 MPa, while (**b**) compares the correlation between actual and predicted germination percentage. (**B**) After-ripened seeds: (**c**) illustrates the germination percentage of *L. campestre* after-ripened seeds at 0 (control), −0.2, −0.4, −0.6, and −0.8 MPa, while (**d**) shows the comparison between the actual and predicted germination percentage, illustrating the linear relationship with the R^2^ and y values. The polyethylene glycol 8000 (Sigma-Aldrich, Germany) solutions were prepared using a method developed by Michel [39], and the germination percentages were determined following the technique outlined by Alvarado and Bradford [40]. Probit regression analysis was performed to determine the best fit between the actual and predicted germination values. The analysis processes the data to identify the maximum germination value at the earliest instance (based on the actual germination values). Each drought treatment consisted of *n* = 3 × 25 samples, and the germination test was conducted over a 4-week period. The actual and predicted germination percentages demonstrate a strong resemblance when the R^2^ value approaches 1. Germination is recorded when the radicle protrusion (2 mm) is visible.

**Figure 5 plants-14-01517-f005:**
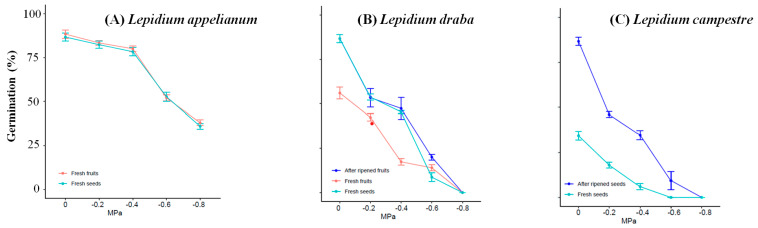
Comparison of germination percentages between the indehiscent fruit-producing *L. appelianum* and *L. draba*, and the dehiscent fruit-producing *L. campestre*. (**A**) illustrates the germination percentage of *L. appelianum* fresh seeds and fruits under different drought stress treatments, while (**B**) displays the germination percentage of *L. draba* fresh seeds and fruits, as well as after-ripened seeds. (**C**) shows the germination percentage of *L. campestre* fresh and after-ripened seeds under different drought stress treatments. The polyethylene glycol 8000 (Sigma-Aldrich, Germany) solutions were prepared using a method developed by Michel [39], and the germination percentages were determined based on the technique outlined by Alvarado and Bradford [40]. Each drought treatment was conducted in triplicate (*n* = 3) with 25 samples, and the germination test was carried out over a 4-week period. Germination is recorded when the radicle protrusion (2 mm) is visible.

## Data Availability

The original contributions presented in this study are included in the article. Further inquiries can be directed to the corresponding author.

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
