# Peer review of "More than Just a Shell: Indehiscent Fruits Drive Drought-Tolerant Germination in Invasive Lepidium Species"

_plants, 2025, doi:10.3390/plants14101517_

Round 1

Reviewer 1 Report

Comments and Suggestions for Authors

This study aims to evaluate the drought stress tolerance of germinating seeds from the invasive, indehiscent fruit-producing Lepidium species, specifically Lepidium appelianum, Lepidium draba, and the dehiscent fruit-producing L. campestre.

The authors focus on investigating how polyethylene glycol (PEG) induced drought stress affects seed and fruit germination in these three invasive weed species: L. appelianum, L. draba, and L. campestre. Gaining an understanding of their drought stress tolerance could provide essential insights for developing strategies to manage invasive species, ultimately enhancing agricultural productivity and biodiversity. Given the native distribution range of these species, it can be hypothesized that L. appelianum and L. draba exhibit greater drought stress tolerance than L. campestre.

The paper presents novel findings.

However, several areas require improvement:

  1. The sections on the dormancy characterization and deospore types of the three Lepidium species, as well as the treatments for releasing dormancy, exhibit repetitive content. Please consider rewriting these sections.
  2. The "Data Analysis" paragraph should be expanded for thorough interpretation of the results. Consider discussing the statistical models employed, significance levels, and any considerations for data reliability and validity.
  3. The discussion should begin by focusing on the researcher's own results before addressing broader issues within the field. This structure will allow for a more meaningful connection between the presented findings and the existing global challenges in the relevant area of study.
  4. Please elaborate on your results by incorporating explanations at the molecular level i.e. if possible to discuss the biochemical processes involved, the role of specific genes or pathways, and any relevant molecular interactions that contribute to the observed phenomena.
  5. It is essential to compare your findings with those from similar studies in the literature. Highlight the similarities and differences in methodologies, results, and interpretations to position your research within the context of existing knowledge.
  6. In the conclusion, it's important to clearly emphasize the novelty and significance of your research findings. Revise this section to articulate the main contributions of your work, making it explicit how it advances the field.
  7. The literature survey should be expanded by incorporating 4-6 additional relevant studies from this journal.

Author Response

Detailed response to reviewer #1

Comments and Suggestions for Authors

This study aims to evaluate the drought stress tolerance of germinating seeds from the invasive, indehiscent fruit-producing Lepidium species, specifically Lepidium appelianum, Lepidium draba, and the dehiscent fruit-producing L. campestre.

The authors focus on investigating how polyethylene glycol (PEG) induced drought stress affects seed and fruit germination in these three invasive weed species: L. appelianum, L. draba, and L. campestre. Gaining an understanding of their drought stress tolerance could provide essential insights for developing strategies to manage invasive species, ultimately enhancing agricultural productivity and biodiversity. Given the native distribution range of these species, it can be hypothesized that L. appelianum and L. draba exhibit greater drought stress tolerance than L. campestre.

The paper presents novel findings.

General

Dear reviewer, thank you for your insightful feedback. We have revised the manuscript in accordance with your suggestions and have restructured the outline to align with the format preferred by the journal (Plants). The manuscript is now organized as follows: Introduction, Results, Discussion, Materials and Methods, and Conclusions. We appreciate your consideration of these changes as you review the updated version.

However, several areas require improvement:

  1. The sections on the dormancy characterization and deospore types of the three Lepidium species, as well as the treatments for releasing dormancy, exhibit repetitive content. Please consider rewriting these sections.

We have addressed your comments and made the necessary corrections in the revised manuscript. Please find the updated version of the manuscript for your review.

  1. The "Data Analysis" paragraph should be expanded for thorough interpretation of the results. Consider discussing the statistical models employed, significance levels, and any considerations for data reliability and validity.

We have made the necessary corrections in the revised manuscript based on your suggestions. Please find the updated manuscript for your consideration.

  1. The discussion should begin by focusing on the researcher's own results before addressing broader issues within the field. This structure will allow for a more meaningful connection between the presented findings and the existing global challenges in the relevant area of study.

Thank you for your thoughtful comments. In response, we have added a new section (3.1) that highlights the main findings of our study and compares them with relevant literature. We believe this enhancement provides a clearer context and strengthens the overall discussion.

3.1. Indehiscent Fruits Drive Enhanced Drought Tolerance and Invasive Potential in Lepidium

The germination responses of the three different species, L. appelianum, L. draba, and L. campestre, to drought stress highlight significant variances in drought tolerance related to their fruit and dispersal types. L. appelianum and L. draba, both producing indehiscent fruits, demonstrated robust germination rates even under varying drought conditions, with L. appelianum achieving an impressive 88.3% germination without drought stress and still maintaining 38% viability at considerable drought stress (-0.8 MPa). This adaptability in survival and reproduction, particularly in environments with limited water availability, provides L. appelianum and L. draba a competitive edge over L. campestre, which exhibited severe reductions in germination success under similar drought stress. The capacity of indehiscent fruits to retain moisture and possibly enhance seedling establishment allows these species to spread effectively in arid regions, raising concerns about their invasiveness and the ecological implications for the native flora. Moreover, the findings align with existing literature highlighting the crucial role of seed and fruit morphology in shaping drought resilience. For instance, Kintl et al. [43] found that morphological traits significantly influence germination rates under drought stress across different clover species. This reinforces the idea that the indehiscent fruits of L. appelianum and L. draba provide not only a survival advantage but also enhance their capacity to outcompete native species, facilitating their establishment in disturbed environments. Such behaviour, driven by the evolutionary advantages conferred by fruit type, highlights the importance of understanding interspecies dynamics in the context of climate change and resource scarcity.

Other studies emphasize the role of genetic traits and environmental factors in drought tolerance, highlighting the invasive potential of indehiscent fruit-bearing species like L. appelianum and L. draba. Mehmandar et al. [44] found that drought stress negatively impacted the physiological traits of Iranian melon genotypes, mirroring our findings of reduced germination rates for L. campestre. In contrast, L. appelianum and L. draba exhibited higher germination rates under drought conditions, suggesting that their indehiscent fruit morphology aids in seed retention and moisture conservation, enabling these species to thrive and compete effectively against L. campestre. Additionally, Li et al. [45] and Licaj et al. [46] highlighted that environmental factors like temperature and photoperiod are crucial for drought resistance, allowing L. appelianum and L. draba to adapt to changing climates. Furthermore, key genes linked to drought tolerance, as noted by Lu et al. [47], may facilitate breeding efforts for these invasive species. Further, the dormancy patterns exhibited by L. draba further complicate the implications of these findings, as the species displayed significant dormancy in its fresh fruits, influencing initial germination rates. However, after-ripening improved germination significantly, indicating that environmental stress cues can enhance the viability of seeds over time. Such mechanisms suggest that L. draba and L. appelianum are well-equipped to thrive under fluctuating climate conditions, effectively positioning them as models for studying invasive traits and resilience strategies. Understanding the distinctions of fruit type in relation to drought resilience not only contributes to our knowledge of plant ecology but also raises critical questions about the management of invasive species in increasingly arid landscapes.

  1. Please elaborate on your results by incorporating explanations at the molecular level i.e. if possible to discuss the biochemical processes involved, the role of specific genes or pathways, and any relevant molecular interactions that contribute to the observed phenomena.

Thank you for your thoughtful comment. However, the primary focus of the present study is on the eco-physiological responses of the species under drought stress conditions. It is important to note that a single study cannot address all research questions, especially those spanning from physiological to molecular levels.

We would like to point out that we have discussed the genetic relationships among the study species in section 3.2, titled "Implications of Indehiscent Fruit Evolution for Drought Stress Adaptation in Lepidium."  Thank you for your understanding.

  1. It is essential to compare your findings with those from similar studies in the literature. Highlight the similarities and differences in methodologies, results, and interpretations to position your research within the context of existing knowledge.

In response to your feedback, we have introduced a new topic in the first section of the discussion, where we compare our results with relevant findings from other studies. We believe this addition strengthens our analysis and provides a more comprehensive context for our results.

3.1. Indehiscent Fruits Drive Enhanced Drought Tolerance and Invasive Potential in Lepidium

The germination responses of the three different species, L. appelianum, L. draba, and L. campestre, to drought stress highlight significant variances in drought tolerance related to their fruit and dispersal types. L. appelianum and L. draba, both producing indehiscent fruits, demonstrated robust germination rates even under varying drought conditions, with L. appelianum achieving an impressive 88.3% germination without drought stress and still maintaining 38% viability at considerable drought stress (-0.8 MPa). This adaptability in survival and reproduction, particularly in environments with limited water availability, provides L. appelianum and L. draba a competitive edge over L. campestre, which exhibited severe reductions in germination success under similar drought stress. The capacity of indehiscent fruits to retain moisture and possibly enhance seedling establishment allows these species to spread effectively in arid regions, raising concerns about their invasiveness and the ecological implications for the native flora. Moreover, the findings align with existing literature highlighting the crucial role of seed and fruit morphology in shaping drought resilience. For instance, Kintl et al. [43] found that morphological traits significantly influence germination rates under drought stress across different clover species. This reinforces the idea that the indehiscent fruits of L. appelianum and L. draba provide not only a survival advantage but also enhance their capacity to outcompete native species, facilitating their establishment in disturbed environments. Such behaviour, driven by the evolutionary advantages conferred by fruit type, highlights the importance of understanding interspecies dynamics in the context of climate change and resource scarcity.

Other studies emphasize the role of genetic traits and environmental factors in drought tolerance, highlighting the invasive potential of indehiscent fruit-bearing species like L. appelianum and L. draba. Mehmandar et al. [44] found that drought stress negatively impacted the physiological traits of Iranian melon genotypes, mirroring our findings of reduced germination rates for L. campestre. In contrast, L. appelianum and L. draba exhibited higher germination rates under drought conditions, suggesting that their indehiscent fruit morphology aids in seed retention and moisture conservation, enabling these species to thrive and compete effectively against L. campestre. Additionally, Li et al. [45] and Licaj et al. [46] highlighted that environmental factors like temperature and photoperiod are crucial for drought resistance, allowing L. appelianum and L. draba to adapt to changing climates. Furthermore, key genes linked to drought tolerance, as noted by Lu et al. [47], may facilitate breeding efforts for these invasive species. Further, the dormancy patterns exhibited by L. draba further complicate the implications of these findings, as the species displayed significant dormancy in its fresh fruits, influencing initial germination rates. However, after-ripening improved germination significantly, indicating that environmental stress cues can enhance the viability of seeds over time. Such mechanisms suggest that L. draba and L. appelianum are well-equipped to thrive under fluctuating climate conditions, effectively positioning them as models for studying invasive traits and resilience strategies. Understanding the distinctions of fruit type in relation to drought resilience not only contributes to our knowledge of plant ecology but also raises critical questions about the management of invasive species in increasingly arid landscapes.

  1. In the conclusion, it's important to clearly emphasize the novelty and significance of your research findings. Revise this section to articulate the main contributions of your work, making it explicit how it advances the field.

We have revised the relevant section of the manuscript as follows:

  1. Conclusions

This study has successfully assessed the drought stress tolerance of germinating seeds and fruits from three Lepidium species: L. appelianum, L. draba, and L. campestre. The results demonstrate that L. appelianum exhibits the highest tolerance to drought, germinating effectively across all drought stress treatments. Lepidium draba shows moderate tolerance, with limited germination under the most severe conditions, while L. campestre failed to germinate in the harsher drought environments. These findings highlight significant differences in drought tolerance, which may be linked to the climatic conditions of each species’ native distribution. The pronounced drought tolerance of L. appelianum and L. draba raises important ecological concerns regarding their potential invasiveness, particularly in the context of climate change. Given their ability to thrive under drought conditions, it is critical to formulate targeted management strategies to control these invasive species effectively. For future research, it is recommended to investigate the underlying physiological mechanisms that contribute to their drought tolerance and to assess their interactions with native plant species. Such studies will provide deeper insights into the ecological impacts of these invasive Lepidium species and aid in developing effective conservation and management practices.

  1. The literature survey should be expanded by incorporating 4-6 additional relevant studies from this journal.

Thank you for your valuable comments and suggestions. In response, we have incorporated five additional references that are directly relevant to our work. We believe these new sources enhance the depth and context of our study.

  1. Kintl, A.; Huˇnady, I.; Vymyslický, T.; Ondrisková, V.; Hammerschmiedt, T.; Brtnický, M.; Elbl, J. Effect of Seed Coating and PEG-Induced Drought on the Germination Capacity of Five Clover Crops. Plants 2021, 10, 724.
  2. Mehmandar, M.N.; Rasouli, F.; Giglou, M.T.; Zahedi, S.M.; Hassanpouraghdam, M.B.; Aazami, M.A.; Tajaragh, R.P.; Ryant, P.; Mlcek, J. Polyethylene Glycol and Sorbitol-Mediated In Vitro Screening for Drought Stress as an Efficient and Rapid Tool to Reach the Tolerant Cucumis melo Genotypes. Plants 2023, 12, 870.
  3. Li, S.; Yan, N.; Tanveer, M.; Zhao, Z.; Jiang, L.;Wang, H. Seed Germination Ecology of the Medicinal Plant Peganum harmala (Zygophyllaceae). Plants 2023, 12, 2660.
  4. Licaj, I.; Fiorillo, A.; Di Meo, M.C.; Varricchio, E.; Rocco, M. Effect of Polyethylene Glycol-Simulated Drought Stress on Stomatal Opening in “Modern” and “Ancient” Wheat Varieties. Plants 2024, 13, 1575.
  5. Lu, G.; Tian, Z.; Chen, P.; Liang, Z.; Zeng, X.; Zhao, Y.; Li, C.; Yan, T.; Hang, Q.; Jiang, L. Comprehensive Morphological and Molecular Insights into Drought Tolerance Variation at Germination Stage in Brassica napus Accessions. Plants 2024, 13, 3296.

Reviewer 2 Report

Comments and Suggestions for Authors

This study suggests that germinating seeds and fruits of L. appelianum demonstrate the strongest tolerance to drought stress, while L. draba exhibits moderate tolerance, and L. campestre seeds display the least tolerance to drought stress. These results provide important support for the control of alien species under climate change. However, there are still minor problems that need to be modified.

Q1: L30: keyword: “Climate change” , do not seem crucial in this article. It is suggested deleted.

Q2: Line 46-52: This part seems a little disconnected from the content of this paragraph

Q 3: Line 85-87: The basis for putting forward this hypothesis is not sufficient.

Q 4: Pictures 1-4 and 5-6 seem to be of different styles. It is suggested to unify them.

Q 5: “4.2. Native distribution areas of indehiscent-fruited Lepidium species suggest drought stress tolerance” This part is an inference and not direct evidence. The author should pay attention to the choice of words and not use the absolute tone; instead, "possible, may" should be used

Author Response

Detailed response to reviewer #2

Comments and Suggestions for Authors

This study suggests that germinating seeds and fruits of L. appelianum demonstrate the strongest tolerance to drought stress, while L. draba exhibits moderate tolerance, and L. campestre seeds display the least tolerance to drought stress. These results provide important support for the control of alien species under climate change. However, there are still minor problems that need to be modified.

General

Dear reviewer, thank you for your insightful feedback. We have revised the manuscript in accordance with your suggestions and have restructured the outline to align with the format preferred by the journal (Plants). The manuscript is now organized as follows: Introduction, Results, Discussion, Materials and Methods, and Conclusions. We appreciate your consideration of these changes as you review the updated version.

Q1: L30: keyword: “Climate change”, do not seem crucial in this article. It is suggested deleted.

Revisions have been made!

Q2: Line 46-52: This part seems a little disconnected from the content of this paragraph

We have made some minor modifications to enhance clarity and coherence, ensuring that the paragraph reads well now.

“……..Our study focuses on closely related invasive indehiscent fruit producing L. appelianum and L. draba and dehiscent fruit producing…….”

Q 3: Line 85-87: The basis for putting forward this hypothesis is not sufficient.

Corrections have been made; please find the revisions at the end of the introduction section.

Given the significant impact of invasive weeds on agriculture and wildlife diversity, understanding the ecological significance of dehiscent and indehiscent fruits is crucial for enhancing agricultural productivity and conserving biodiversity. Insights into the germination behaviours and adaptive strategies of these diaspores can inform effective management plans aimed at mitigating their harmful effects on ecosystems and farming practices.

Q 4: Pictures 1-4 and 5-6 seem to be of different styles. It is suggested to unify them.

We appreciate your feedback but would like to clarify our understanding of your point. If you are referring to Figure 1, the first four images depict the indehiscent fruit-producing species L. appelianum and L. draba, while images five and six represent the dehiscent fruit-producing species L. campestre. It is expected to observe variations in style between the dispersal types of the study species, given that they possess different dispersal units (dehiscent vs. indehiscent) and have distinct visual characteristics associated with these units.

Q 5: “4.2. Native distribution areas of indehiscent-fruited Lepidium species suggest drought stress tolerance” This part is an inference and not direct evidence. The author should pay attention to the choice of words and not use the absolute tone; instead, "possible, may" should be used

We have revised the section by incorporating the word “could,” and it now reads as follows:

“2.3. Native distribution areas of indehiscent-fruited Lepidium species could suggest drought stress tolerance”

Reviewer 3 Report

Comments and Suggestions for Authors

I have found the present manuscript is interesting and has great application value.   Overall, the text of the manuscript is over a currently important topic, the subject meets well with the journal's scope and statistical analyses applied were, in my opinion, appropriate and proper. But there are some areas that require revision for clarity. I hope my comments are helpful to the authors in revising their manuscript.

1. Figure 1 contains unclear annotations (e.g., "Reagido arabus" in panel B appears to be a typo; it should likely be corrected to Lepidium draba).Legends for Figures 2–5 mention "predicted germination" but lack details on the prediction model (e.g., whether linear regression or other statistical methods were used).

2.References exhibit formatting inconsistencies (e.g., journal names are inconsistently abbreviated; some entries lack volume numbers or page ranges). Examples:

Reference 1 (Zahedifar & Zohrabi, 2016) lacks a volume number.

Reference 28 (Mohammed et al., 2019) fails to italicize the journal name (Weed Science).

3."MPa" is not italicized (correct format: MPa).

4.Repetitive descriptions of data in the Results section (e.g., "L. appelianum fresh seeds and fruits germinated under all drought stress treatments" appears redundantly in the Abstract and Results). Overly complex sentence structures (e.g., Discussion section 4.1: "The shift from dehiscent to indehiscent in fruit could potentially be understood..." could be simplified for clarity).

5.Spelling variations: "after-ripened" vs. "after-ripend" (Figure 3).

6.Species naming inconsistency: "L. áraba" in the Abstract vs. "L. draba" in the main text; standardize to align with accepted nomenclature.

7.Single Seed Source: All seeds were sourced from cultivated plants in a German botanical garden. This neglects potential genetic variation across geographic populations, which may limit the generalizability of the conclusions.

8.Ecological Relevance of PEG Simulation: The study does not discuss how PEG-induced osmotic stress correlates with real-world soil drought dynamics (e.g., fluctuating soil moisture levels vs. static PEG conditions).

9,Vague Statistical Methods: While "one-way ANOVA" and "Tukey’s HSD" are mentioned, the manuscript omits critical details (e.g., tests for normality and homogeneity of variances).

10. Lack of Model Interpretation: Linear models for predicted germination rates (e.g., R²=0.8018) are presented without biological context or justification for their use.

11.The emphasis on indehiscence as a driver of drought tolerance overlooks potential confounding traits (e.g., seed mucilage, size, or microbial interactions). Multivariate analyses are needed to isolate the role of fruit type.

Author Response

Detailed response to reviewer #3

Comments and Suggestions for Authors

I have found the present manuscript is interesting and has great application value.   Overall, the text of the manuscript is over a currently important topic, the subject meets well with the journal's scope and statistical analyses applied were, in my opinion, appropriate and proper. But there are some areas that require revision for clarity. I hope my comments are helpful to the authors in revising their manuscript.

General

Dear reviewer, thank you for your insightful feedback. We have revised the manuscript in accordance with your suggestions and have restructured the outline to align with the format preferred by the journal (Plants). The manuscript is now organized as follows: Introduction, Results, Discussion, Materials and Methods, and Conclusions. We appreciate your consideration of these changes as you review the updated version.

  1. Figure 1 contains unclear annotations (e.g., "Reagido arabus" in panel B appears to be a typo; it should likely be corrected to Lepidium draba).Legends for Figures 2–5 mention "predicted germination" but lack details on the prediction model (e.g., whether linear regression or other statistical methods were used).

We have revised the data analysis section, as well as the legends for Figures 2-4, by adding the following information:

Probit regression analysis was performed to determine the best fit between the actual and predicted germination values. The analysis processes the data to identify the maximum germination value at the earliest occurrence based on the observed germination values.”

2.References exhibit formatting inconsistencies (e.g., journal names are inconsistently abbreviated; some entries lack volume numbers or page ranges). Examples:

The references have been thoroughly revised throughout the manuscript.

Reference 1 (Zahedifar & Zohrabi, 2016) lacks a volume number.

Zahedifar, M.; Zohrabi, S. Germination and seedling characteristics of drought-stressed corn seedasinfluenced by seed priming with potassium nano-chelate and sulfate fertilizers. Acta Agric.Slov. 2016, 107, 113 – 128. The number 107 signifies the volume number.

Reference 28 (Mohammed et al., 2019) fails to italicize the journal name (Weed Science).

Corrections have been made; thank you for your input. We have revised the reference section throughout.

“Mohammed, S.; Turečková, V.; Tarkowská, D.; Strnad, M.; Mummenhoff, K.; Leubner-Metzger, G. Pericarp-mediated chemical dormancy controls the fruit germination of the invasive hoary cress (Lepidium draba), but not of hairy whitetop (Lepidium appelianum). Weed Sci. 2019, 67, 560571.”

3."MPa" is not italicized (correct format: MPa).

Thank you for your feedback. Corrections have been made throughout the manuscript.

4.Repetitive descriptions of data in the Results section (e.g., "L. appelianum fresh seeds and fruits germinated under all drought stress treatments" appears redundantly in the Abstract and Results). Overly complex sentence structures (e.g., Discussion section 4.1: "The shift from dehiscent to indehiscent in fruit could potentially be understood..." could be simplified for clarity).

We have revised the abstract and results sections to eliminate redundant wording. Additionally, we have updated the title of the discussion section 3.2 to the following:

3.2. Implications of Indehiscent Fruit Evolution for Drought Adaptation in Lepidium

5.Spelling variations: "after-ripened" vs. "after-ripend" (Figure 3).

Thank you for your careful review. We have corrected the spelling error as noted.

6.Species naming inconsistency: "L. áraba" in the Abstract vs. "L. draba" in the main text; standardize to align with accepted nomenclature.

We did not find the species name "L. áraba" in the abstract section. Instead, we observed that the correct species name "L. draba" is consistently used throughout the manuscript.

7.Single Seed Source: All seeds were sourced from cultivated plants in a German botanical garden. This neglects potential genetic variation across geographic populations, which may limit the generalizability of the conclusions.

Thank you for your insightful comment. The seeds used in our study were sourced from their native distribution sites. Please refer to Section 4.1 of the Materials and Methods for details on seed sources. We conducted mass propagation in the botanical garden of Osnabrueck University (Germany) to ensure a sufficient quantity of seeds. It is important to emphasize that the sources are from their native distribution areas, and our comparisons are made within the context of these regions and their potential invasiveness.

8.Ecological Relevance of PEG Simulation: The study does not discuss how PEG-induced osmotic stress correlates with real-world soil drought dynamics (e.g., fluctuating soil moisture levels vs. static PEG conditions).

We have taken your suggestions into account and added a new topic in the discussion section that explores the ecological relevance of PEG for simulating drought stress. This addition aims to provide a deeper understanding of the implications and applications of our findings in ecological contexts.

3.6. The Broad Ecological Relevance of PEG in Drought Stress Simulation

Polyethylene glycol (PEG) has emerged as an essential compound in ecological research for simulating drought stress conditions during the critical stages of seed germination and seedling growth [7]. By inducing controlled osmotic stress in laboratory settings, PEG allows researchers to maintain consistent water potential while observing how various plant species respond to water scarcity [13]. This capability is vital for understanding the impacts of drought on seed vigour, germination rates, and overall seedling health. Additionally, PEG facilitates the exploration of plant physiological and biochemical responses, which can be crucial for developing resilient crop varieties and enhancing agricultural productivity in an era marked by climate variability [7]. Beyond its agricultural applications, PEG in the context of inducing controlled drought stress plays a significant role in studying the ecological dynamics of invasive plant species [33]. By examining how controlled drought stress via PEG influences seed traits and germination strategies, researchers can uncover the mechanisms that enable certain species to thrive under water-limiting conditions [34]. Such insights are critical for devising effective management strategies aimed at controlling the spread of invasive species and protecting native ecosystems [35]. Furthermore, PEG's simulation of drought conditions has proven beneficial in elucidating the interactions between seed mucilage, water absorption, and seed dispersal, contributing to a deeper understanding of plant survival strategies in increasingly unpredictable environments [38]. Through these multifaceted studies, PEG serves as both a vital research tool and a key component in addressing the ecological challenges posed by drought stress.

9,Vague Statistical Methods: While "one-way ANOVA" and "Tukey’s HSD" are mentioned, the manuscript omits critical details (e.g., tests for normality and homogeneity of variances).

Thank you for your insightful comment. We have revised the data analysis section in accordance with your suggestions:

4.7. Data analysis

One-way ANOVA was conducted to analyse the association between the drought stress treatments within and between the species. Data were analysed using a one-way ANOVA to assess differences among groups, followed by post hoc comparisons using Tukey's honest significant difference test. The assumption of homogeneity of variances was evaluated to ensure the validity of the results. The rejection threshold for all analyses was P < 0.05. Probit regression analysis was performed to determine the best fit between the actual and predicted germination values. The analysis processes the data to identify the maximum germination value at the earliest occurrence based on the observed germination values. The results were then visually represented using R version 4.3.2 (The R Foundation for Statistical Computing).

  1. Lack of Model Interpretation: Linear models for predicted germination rates (e.g., R²=0.8018) are presented without biological context or justification for their use.

We have included the regression model utilized in the study and made modifications to the data analysis section. Additionally, we have updated the legends for Figures 2-5 by adding the following information:

 Probit regression analysis was performed to determine the best fit between the actual and predicted germination values. The analysis processes the data to identify the maximum germination value at the earliest instance (based on the actual germination values).”

11.The emphasis on indehiscence as a driver of drought tolerance overlooks potential confounding traits (e.g., seed mucilage, size, or microbial interactions). Multivariate analyses are needed to isolate the role of fruit type.

Thank you for your insightful comment. We believe that we have thoroughly studied and gained a clear understanding of other variables, such as seed mucilage, seed size and microbial interactions, among others, as discussed in our other manuscripts, also some of them cited here.

  1. Mohammed, ; Mummenhoff, K. Functional divergence exists in mucilage-mediated seed dispersal, but not in germination of myxospermic Lepidium campestre and Lepidium draba (Brassicaceae). Acta Oecologica 2024, 125, 104042.
  2. Mohammed, S.; Steinbrecher, T.; Leubner-Metzger, G.; Mummenhoff, K. Differential Primary Seed and Fruit Dispersal Mechanisms and Dispersal Biomechanics in Invasive Dehiscent and Indehiscent-Fruited LepidiumPlants 2025, 14, 446.
  3. Mohammed S, Turečková V, Tarkowská D, Strnad M, Mummenhoff K, Leubner-Metzger G. Pericarp-mediated chemical dormancy controls the fruit germination of the invasive hoary cress (Lepidium draba), but not of hairy whitetop (Lepidium appelianum). Weed Science. 2019;67(5):560-571. doi:10.1017/wsc.2019.33.
  4. Mohammed S, Bhattacharya S, Gesing MA, Klupsch K, Theißen G, Mummenhoff K, Müller C. Morphologically and physiologically diverse fruits of two Lepidium species differ in allocation of glucosinolates into immature and mature seed and pericarp. PLoS One. 2020 Aug 25;15(8):e0227528. doi: 10.1371/journal.pone.0227528. PMID: 32841235; PMCID: PMC7447065.
  5. Mohammed,S., Samik Bhattacharya,S.,  Mummenhoff, K. (2019). Dead or Alive: Simple, Nondestructive, and Predictive Monitoring of Seedbanks. Trends in Plant Sciences 24:783–784.
  6. Mohammed, S; Mummenhoff, K. Germination under Temperature Stress Facilitates Invasion in Indehiscent Lepidium Species, Agriculture, 2025 (In press)

Based on our extensive experience and the numerous experiments we have conducted regarding invasive species, particularly Lepidium, we believe that the findings of this study are robust and significant. We are confident in our ability to analyse and draw conclusions about the impact of indehiscent fruit-producing Lepidium species in relation to drought stress tolerance. In response to your comments, we have incorporated relevant information into Section 3.6 of the discussion:

…… Furthermore, PEG's simulation of drought conditions has proven beneficial in elucidating the interactions between seed mucilage, water absorption, and seed dispersal, contributing to a deeper understanding of plant survival strategies in increasingly unpredictable environments [38]. Through these multifaceted studies, PEG serves as both a vital research tool and a key component in addressing the ecological challenges posed by drought stress.

Round 2

Reviewer 3 Report

Comments and Suggestions for Authors I think the manuscript has been sufficiently improved to warrant publication in Plants.